# Interfacial Interaction between Functionalization of Polysulfone Membrane Materials and Protein Adsorption

**DOI:** 10.3390/polym16121637

**Published:** 2024-06-10

**Authors:** Sheng Yan, Yunren Qiu

**Affiliations:** College of Chemistry and Chemical Engineering, Central South University, Changsha 410083, China; ysheng326@163.com

**Keywords:** cell adhesion, molecular dynamic simulation, polysulfone, protein

## Abstract

This study that modified polysulfone membranes with different end-group chemical functionalities were prepared using chemical synthesis methods and experimentally characterized. The molecular dynamics (MD) method were used to discuss the adsorption mechanism of proteins on functionalized modified polysulfone membrane materials from a molecular perspective, revealing the interactions between different functionalized membrane surfaces and protein adsorption. Theoretical analysis combined with basic experiments and MD simulations were used to explore the orientation and spatial conformational changes of protein adsorption at the molecular level. The results show that BSA exhibits different variability and adsorption characteristics on membranes with different functional group modifications. On hydrophobic membrane surfaces, BSA shows the least stable configuration stability, making it prone to nonspecific structural changes. In addition, surface charge effects lead to electrostatic repulsion for BSA and reduce the protein adsorption sites. These MD simulation results are consistent with experimental findings, providing new design ideas and support for modifying blood-compatible membrane materials.

## 1. Introduction

Polysulfone (PSf) is a kind of high polymer containing aromatic ring, sulfonyl group and aromatic ether bond in the main chain [1]. It exhibits excellent mechanical properties and outstanding oxidation and hydrolysis resistance [2,3]. The blood dialysis membrane made from sulfone materials has a good clearance rate for middle molecule toxins and is widely utilized [4,5]. However, due to the high number of aromatic rings in the material itself, it has poor hydrophilicity, leading to protein adsorption. Non-specific protein adsorption can cause coagulation, while protein adsorption on the surface of the material can promote blood cell adhesion, resulting in thrombosis [6]. 

Therefore, there is an urgent need to understand the principles of protein adsorption as well as how proteins maintain stability on compatible materials for a long time [7,8]. Previous studies on protein adsorption have generated a large amount of experimental data research. However, due to limitations in experimental methods, obtaining information about the orientations, conformations, and interactions of biological molecules such as proteins during their adsorption process is challenging [9]. It is impossible to intuitively demonstrate the process of protein adsorption at a molecular level [10].

When blood in contact with a membrane material, the protein will quickly adsorb to the surface of the membrane material, which will lead to platelet adhesion, aggregation, coagulation, and eventually the formation of blood clots. Therefore, in order to prevent thrombosis after blood contacts the membrane, it is necessary to block the adsorption of proteins in plasma on the surface of the membrane. 

This paper utilizes the molecular dynamics (MD) method to analyze protein adsorption on membrane surfaces from a molecular perspective [11,12,13]. Currently, different functional groups are grafted onto PSf membranes in order to improve their anti-protein properties by regulating different chemical functional groups at their ends. The hydrophilicity and blood compatibility of polysulfone membranes are modified from the perspective of functional group grafting. Representative functional groups with different charge properties and hydrophilicity are selected, such as -CH_3_, -OH, -COOH, and -SO_3_H. Then, the polysulfone membranes are grafted with different functional groups through surface chemical reactions, and the hydrophilicity and other properties of the modified membrane materials are tested. We then explored how modified PSf membrane surfaces with -OH, -COOH, and -SO_3_H affect bovine serum albumin (BSA) absorption through basic experiments using the MD method and analyzed their interactions [14,15,16]. This provides a theoretical basis for surface modification of PSf membranes and improving their anti-protein properties. 

## 2. Materials and Methods

### 2.1. Materials

Polysulfone (PSf average Mn: 22,000) were provided by Rhawn. Sodium hydroxide (NaOH,), sodium azide (NaN_3_), sodium iodide (NaI), and sodium borohydride (NaBH_4_) were of analytical grade and were purchased from Sinopharm Group Chemical reagent Co., Ltd., Shanghai, China. Trichloromethyl silane (99%), paraformaldehyde (95%), pyridine sulfur trioxide (98%), and succinic anhydride (99%) were provided by Aladdin Industrial Corporation, Shanghai, China.

### 2.2. Modification of Polysulfone Membrane

Different functional groups, including hydroxyl groups (-OH), carboxyl groups (-COOH), and sulfo groups (-SO_3_H) were introduced onto the PSf membrane surfaces via surface chemical reactions [2].

Preparation of the hydroxylate polysulfone membrane (PSf-OH) was performed as follows: Chloromethylated polysulfone (PSf-Cl) was synthesized through a gentle and controlled process. Then, the PSf-Cl membrane was prepared using the liquid–liquid phase separation method. In order to fabricate a hydroxylated PSf membrane (PSf-OH), the PSf-Cl membrane was immersed in a NaOH aqueous solution at 300 K for approximately 72 h, followed by thorough rinsing with ample ultrapure water.

Preparation of carboxylated polysulfone membrane (PSf-COOH) was performed as follows: The azido-polysulfone (PSf-N_3_) membrane was prepared from the PSf-Cl membrane through an azidation reaction with NaN_3_ and NaI at 333 K for 24 h. Subsequently, the aminated membrane (PSf-NH_2_) was obtained by reacting the azide groups with NaBH_4_ (20 mg/mL aqueous solution) at 333 K for 6 h using the PSf-N_3_ membrane [17]. Furthermore, the carboxylated polysulfone (PSf-COOH) membrane was prepared by reacting the amino groups with succinic anhydride (20 mg/mL alcoholic solution) at 300 K for 24 h using the PSf-NH_2_ membrane [18]. 

Preparation of sulfonated polysulfone membrane (PSf-SO_3_H) was performed as follows: The sulfonated polysulfone membrane (PSf-SO_3_H) was synthesized from the PSf-NH_2_ membrane through a reaction between the amino groups and pyridine sulfur trioxide (20 mg/mL aqueous solution) at 300 K for 24 h [19].

### 2.3. Characterization

The surface chemical compositions were analyzed by X-ray photoelectron spectroscopy (XPS, ESCALAB Xi^+^). Water contact angles were detected by DSA-10 drop shape analyzer according to sessile drop method. Bovine serum albumin (BSA) adhesion tests for the 1 cm × 1 cm membranes were immersed in a phosphate buffer (PBS) solution with a pH of 7.4 for a duration of 24 h. Subsequently, they were removed from the buffer and placed into 1 mg/mL BSA solution at 310 K for a period of 2 h. Following this, the membranes underwent washing with a PBS solution before being immersed in a gently stirred 2 wt.% SDS solution at 310 K to effectively eliminate any adsorbed protein. The concentration of BSA in the SDS washing solution was determined using a UV spectrophotometer set at an absorbance wavelength of 280 nm, and the standard curve line was generated using BSA solutions. 

### 2.4. Construction of Computational Model

The models were established using GROMACS software (Version 2019.6) to correspond with the experimental preparations. Four sets were utilized, including PSf, PSf-OH, PSf-COOH, and PSf-SO_3_H. These models, based on benzene ring molecules, were placed in a custom unit cell and replicated in the x and y directions to construct a supercell for obtaining planar dimensions. The quantity of hydrogen atoms in the terminal groups of PSf, PSf-OH, PSf-COOH, and PSf-SO_3_H was altered to achieve charged surfaces with identical charge density.

Albumin is the most abundant protein in plasma and has a very important role in the blood. Bovine serum albumin (BSA) is a spherical albumin extracted from bovine plasma that is widely used to simulate or test the properties of blood proteins. In molecular dynamics simulations, BSA has been used as an ideal model to simulate or construct proteins, and after many years of research, the complete crystal structure of BSA has been constructed. We chose the bovine serum albumin (BSA) monomer as the protein model. The crystal structure of the BSA is derived from the RCSB protein database and consisted of 1166 amino acid residues. The electric dipole moment of the BSA was 1819.31 D, and the molecular weight of the BSA was 133.28 Kda [20].

The functional properties and state densities of the membrane surfaces (PSf, PSf-OH, PSf-COOH, and PSf-SO_3_H) were calculated using the CASTEP tool from Materials Studio [21]. Four molecular unit cells were established for each type and optimized geometrically using GGA function in CASTEP calculation mode [21]. The optimized unit cells had their (001) surface extracted before setting a vacuum medium on it for calculating surface energy levels using LDA function. Subsequently, the functional properties and energy level densities of the membrane surfaces for PSf, PSf-OH, PSf-COOH, and PSf-SO_3_H molecules were analyzed using the CASTEP analysis mode. 

GROMACS (Version 2019.6)—the software is developed based on C language, which has good force field compatibility, and is suitable for MD simulation of various types of biological macromolecule systems.

The GROMACS software was used for molecular dynamics simulations. Whole atomic force fields were used to construct proteins, and the General Amber force field was used to construct surface molecules. The surface was constructed by arranging 20 polysulfone chains with 36 repeating units, and the end functional groups of the repeating units were replaced by the 4 different functional groups -CH_3_, -COOH, -SO_3_H, and -OH. The constructed surface size was about 20 nm × 12 nm × 12 nm, and the surface had enough space to accommodate protein molecular chains. During all calculations, the protein spacing on the surface was adjusted so that the BSA was in the same spacing position on four different surfaces. The hydrogen atoms of the end groups in the PSf-CH_3_, PSf-OH, PSf-COOH, and PSf-SO_3_H systems were edited to protonate or deprotonate the end groups, respectively, and the charge density of the membrane surface was adjusted to 0.1 C·m^−2^. The heavy atoms of the surface molecules were confined to their positions after the energy was minimized. Firstly, the BSA was placed 0.5 nm away from the surface, more than 70,000 water molecules were added to dissolve the protein, and then 0.1 mol ions were randomly replaced by water molecules. The semi-anisotropic NPT simulation was performed with a surface side length of 5 ns, and the system was balanced through thousands of steps of energy minimization. Finally, at 298 K, the molecular dynamics simulation was performed using a canonical ensemble for 60 ns, the cutoff length of the non-covalent bond interaction was 1.2 nm, and the distant electrostatic interaction was analyzed using a particle grid with 0.1 nm Fourier spacing. 

## 3. Results and Discussion

### 3.1. Surface Characterization

#### 3.1.1. XPS Analysis

By utilizing XPS to analyze the oxygen (O), nitrogen (N), sulfur (S), and carbon (C) content on the surface of PSf, PSf-OH, PSf-COOH, and PSf-SO_3_H membranes, the process of surface modification was further confirmed. As shown in Table 1, compared to PSf, the O content slightly increased (20.54~26.41%) while the C content slightly decreased (76.83~65.29%) for PSf-OH, PSf-COOH, and PSf-SO_3_H. The sulfonated S content significantly increased (2.20~4.99%) after modification. Figure 1 shows three main peaks at 280.0 eV, 530.0 eV, and 180.0 eV which correspond to c1s, O1s, and S2p, respectively. There are also other emission peaks at 400.0 eV, which can be attributed to N1s. The S1s emission peak in the XPS spectrum of PSf -SO_3_H membrane indicates the presence of SO_3_H groups on the membrane surface. Furthermore, from Table 1, it can be seen that N and S contents in original PSf were 0.12% and 2.51%, respectively, which increased to 3.29% and 4.99% after sulfonation, indicating that -SO_3_H is present on the membrane surface.

#### 3.1.2. Water Contact Angle Analysis

The static contact angle is a direct method of evaluating wettability and decreases with increasing hydrophilicity to indicate the hydrophilicity of the film surface. Figure 2 shows the contact angles of the membranes, the contact angles of PSf, PSf-OH, PSf-COOH, and PSf-SO_3_H membranes were 86.2°, 69.5°, 40.1°, and 38.2°, respectively, while the contact angles of PSf and PSf-SO_3_H membranes were significantly reduced to 86.2° to 38.2° after grafting -SO_3_. Therefore, it can be concluded that the hydrophilicity of PSf-OH, PSf-COOH, and PSf-SO_3_H modified film has been greatly improved.

#### 3.1.3. Adsorption of Protein

According to the waterfall sequence coagulation mechanism proposed by Davies, the adsorption of blood protein is the first step of coagulation. After platelet adhesion, various coagulation pathways are further activated, and finally lead to the occurrence of thrombosis genes. The amount of protein adsorbed on the membrane surface is considered to be an important indicator to evaluate the hemocompatibility of the membrane. Our previous work discussed the rationality of using BSA instead of HAS to simulate protein absorption assays. As shown in Figure 3, the adsorption value of BSA on PSf-SO_3_H membrane decreased significantly compared with pure PSf membrane and PSf-OH membrane. As the reaction time increases, the adsorption capacity of PSf-SO_3_H and PSf-COOH membrane decreases further. The significant difference between the sulfonated-modified membrane and PSf membrane indicated that the anti-pollution ability of modified modification membrane to protein is greatly improved. 

### 3.2. Protein Adsorption Mechanism

#### 3.2.1. Protein Configuration Change 

Bovine serum albumin (BSA) is a kind of transport albumin, which is often used in protein adsorption tests, especially it has been used to evaluate the anti-protein adsorption apacity of membrane materials. In addition, BSA molecule has a complete all-atomic structure with high resolution, which is an ideal model protein for detecting protein adsorption correlation simulation methods. In order to study the mechanism of protein adsorption on the surface of membrane materials, BSA was used as the protein model. The model was a BSA monomer, whose crystal structure was derived from the RCSB protein database, with a total of 1166 amino acid residues, molecular weight of 133.28 KDa, electric dipole moment of 1819.31 D, and static charge of −32. During the simulation process, the BSA was placed at the upper 0.5 nm distance from the surface model, as shown in Figure 4. The initial velocity of the atom is given by the Maxwell–Boltzmann distribution at 300 K, the simulated step size is 2 fs, and the coupling time is 0.5 ps.

Figure 5 shows the BSA backbone RMSD during adsorption, that is, the change amplitude of BSA molecular configuration relative to the initial structure [22]. Root mean square deviation (RMSD) refers to the coordinate deviation of the atomic coordinates in the protein main chain molecule relative to the reference structure, which is fitted by the least square method. It is used to indicate that with the change of the simulated time protein configuration, the larger the RMSD value fluctuation range, indicating that the protein configuration is prone to change, and the larger the RMSD value, the higher the probability of protein mutation. The larger the protein mass adsorbed on the surface of the membrane, the larger the fluctuation range of the RMSD value of the protein. Through the experimental characterization, it was found that the protein adsorption amounts was the largest in the PSf system, and the protein adsorption amounts was the smallest in the PSf-SO_3_H system. The simulation showed that the RMSD value of proteins in PSf system had the largest fluctuation range, and the RMSD value of proteins in PSf-SO_3_H system had the smallest fluctuation range. The simulation results are in agreement with the experimental characterization.

Figure 6 shows the RMSF of each residue in the final stage of BSA molecule simulation, representing the stability of each residue [23]. The RMSF of BSA residues in the PSf-OS_3_H system remains at a relatively stable and low level, indicating that the continuous electrostatic attraction on the surface of PSf-OS_3_H plays a role in fixing the configuration of BSA, thus maintaining relative stability of the adsorbed BSA peptide chain.

Figure 7 shows the simulated SASA of each residue in the final stage of BSA. Since the molecular structure of macromolecule BSA is flexible, the protein has been folded, and under the folding action, some residues are wrapped in other molecules and cannot directly contact with the membrane surface, so the SASA value of the residues wrapped in the interior is relatively low. Figure 7 simulated the SASA changes of BSA residues in PSf-CH_3_, PSf-OH, PSf-COOH, and PSf-SO_3_H membrane systems. In addition to the PSf-CH_3_ system, the residue SASA of BSA has a large range of changes, indicating that the BSA has a higher surface energy after contact with the PSf membrane. In the PSf-OH, PSf-COOH, and PSf-SO_3_H membrane systems, the surface of the membrane material is too small to cause protein folding, which further indicates that the smaller the surface energy of the membrane surface, the smaller the probability of irreversible folding and denaturation of the protein during the adsorption process with the membrane surface [24].

#### 3.2.2. Adsorption Orientation and Interactive Binding Sites

The adsorption orientation of proteins on the surface of materials determines the efficiency of their biological activity [24]. The orientation angle (θ) is defined as the angle between the dipole moment of the protein and the normal vector, as shown in Figure 8 for the adsorption orientation distribution curve of BSA at the stable stage. The cosine values corresponding to the peak distribution of BSA orientation in PSf, PSf-OH, PSf-COOH, and PSf-SO_3_H systems are all negative, approximately −0.60, −0.56, and −0.11, respectively, with corresponding orientation angles at 96°. In comparison to the PSf system, PSf-OH and PSf-COOH exhibit narrower peaks in their orientation distributions. This suggests that surface adsorption of BSA is subject to repulsive forces in these systems leading to relatively stable adsorption orientations under sustained repulsive forces.

Figure 9 shows that in an electrically neutral system, there is a lack of long-range electrostatic interaction forces [25,26]. In the negatively charged PSf-OH, PSf-COOH, and PSf-SO_3_H systems, BSA molecules are subject to electrostatic repulsion, but adsorbed due to the polarity of BSA molecules themselves, with fewer binding sites. In the positively charged PSf system, BSA polymer molecules are subjected to continuous electrostatic attraction and adsorbed on the surface of the membrane to form more binding sites. Most of the binding sites are amino acid residues, and the rotation angle is larger relative to the initial orientation, which is approximately a vertical flip.

Hydrogen bonding is a polar force and has the function of anchoring protein molecules. Figure 10 shows the number of hydrogen bonds generated between the BSA and the membrane surface. The number of hydrogen bonds in PSf system is the largest, and due to the electrostatic attraction of the material interface, the contact area between BSA and the surface is larger. The bonds in PSf-COOH and PSf-OH systems are mainly from the local electrostatic attraction. The surface of PSf-SO3H did not form hydrogen bonds with BSA, and the number of hydrogen bonds was approximately 0. Due to the weak interaction between surface and protein, the contact area of BSA was small.

#### 3.2.3. Influence of Interface Interaction Energy

The interaction free energy between BSA and the membrane surface is simulated in PSf-CH_3_-, PSf-OH-, PSf-COOH and PSf-SO_3_H systemsd, including the free energy of Van der Waals interaction ($E_{vdw}$) and electrostatic potential energy ($E_{elect}$). The simulation results are shown in Table 2. The results showed that the adsorption of BSA proteins on the surface of PSf-CH_3_, PSf-OH, PSf-COOH, and PSf-SO_3_H grafted membranes is mainly caused by the synergistic force of Van der Waals interaction free energy ($E_{vdw}$) and electrostatic interaction. Through simulation data analysis, the Van der Waals interaction free energy ($E_{vdw}$) is much smaller than the electrostatic potential energy ($E_{elect}$), indicating that the membrane protein interaction mainly relies on the electrostatic interaction; that is, the dominant force of protein adsorption on the surface of the charged material is the electrostatic potential energy. The electrostatic interaction between the BSA protein and PSf-CH_3_ on the positively charged surface is smaller than that between BSA protein and PSf-OH, PSf-COOH, and PSf-SO_3_H on the negatively charged surface. The higher the density of the negative charge on the membrane surface, the higher the electrostatic potential energy ($E_{elect}$) and the higher the electrostatic interaction between the membrane and the protein interface.

## 4. Conclusions

This paper employs the molecular dynamics (MD) method, combined with experimental characterization, to analyze the adsorption of proteins on the surface of sulfone membranes grafted with different functional groups from a molecular perspective. XPS tests have confirmed the successful grafting of -OH, -COOH, and -SO_3_H functional groups onto the PSf membrane surface. Hydrophilicity and protein adsorption experiments have been conducted, leading to the conclusion that PSf membranes grafted with -COOH and -SO_3_H functional groups exhibit good hydrophilicity and resistance to protein adsorption. The MD method is used to investigate the adsorption process of bovine serum albumin (BSA) on modified PSf membrane surfaces and to analyze their interactions. The analysis reveals that BSA exhibits different conformational stability and adsorption characteristics on membranes modified with different functional groups; BSA shows the poorest stability on PSf membrane surfaces but the best stability on PSf-SO_3_H membrane surfaces, indicating minimal non-specific changes in protein structure. Additionally, due to differences in surface charge among membranes with different functional groups, particularly under electrostatic repulsion effects, BSA has very few binding sites on PSf-SO_3_H membranes, making it less prone to adsorb onto these surfaces. The results of MD simulations are consistent with experimental findings and provide theoretical support for modifying blood-compatible membrane materials.

## Figures and Tables

**Figure 1 polymers-16-01637-f001:**
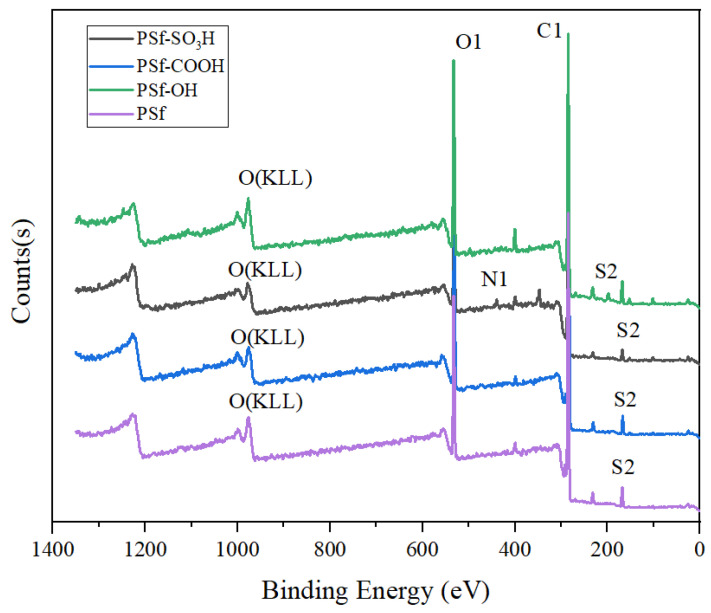
XPS C1s high resolution spectra of the PSf, PSf-OH, PSf-COOH, and PSf-SO_3_H membranes.

**Figure 2 polymers-16-01637-f002:**
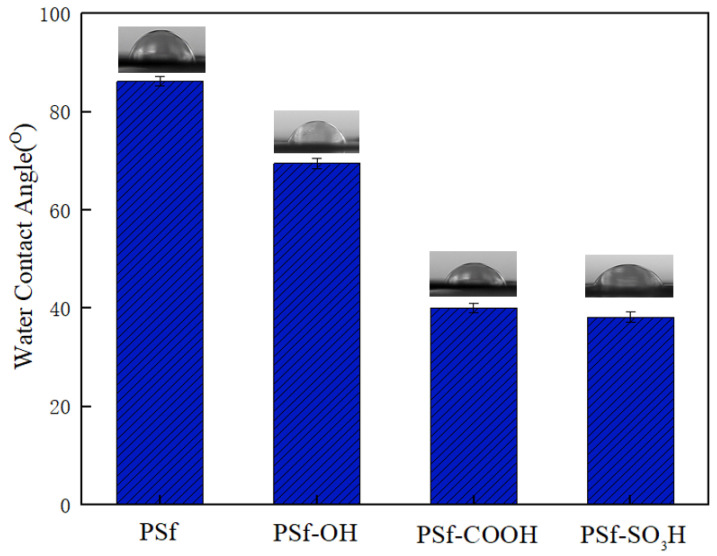
Static water contact angles for different membranes. Data are expressed as the mean ± SD of one independent measurement.

**Figure 3 polymers-16-01637-f003:**
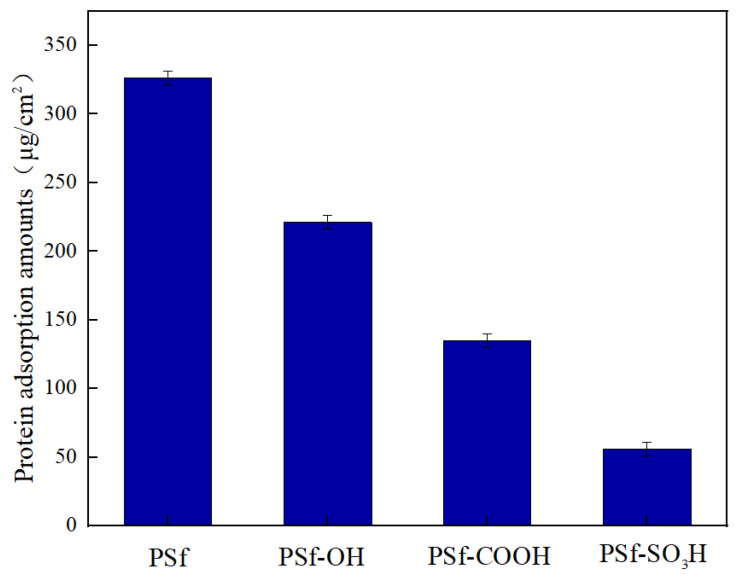
The BSA adsorption on the surfaces of PSf, PSf-OH, PSf-COOH, and PSf-SO_3_H membranes. Data are expressed as the mean ± SD of six independent measurements.

**Figure 4 polymers-16-01637-f004:**
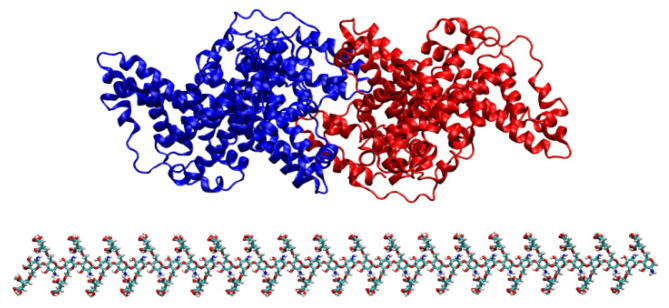
MD simulates the initial system state.

**Figure 5 polymers-16-01637-f005:**
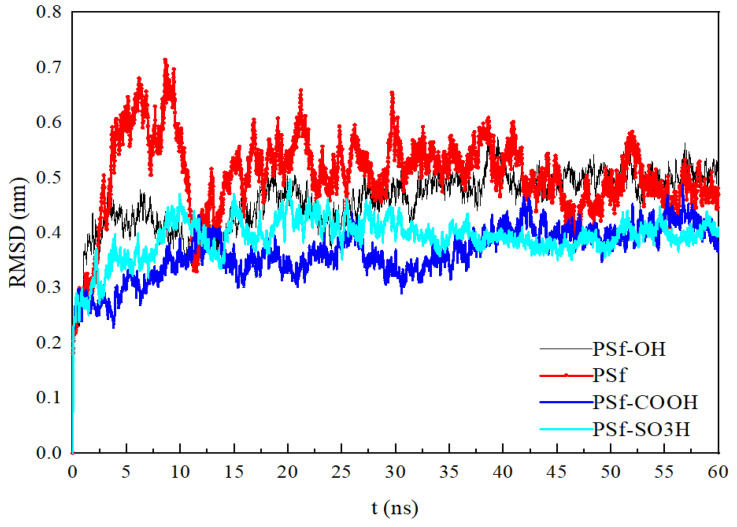
Main chain RMSD of BSA during adsorption.

**Figure 6 polymers-16-01637-f006:**
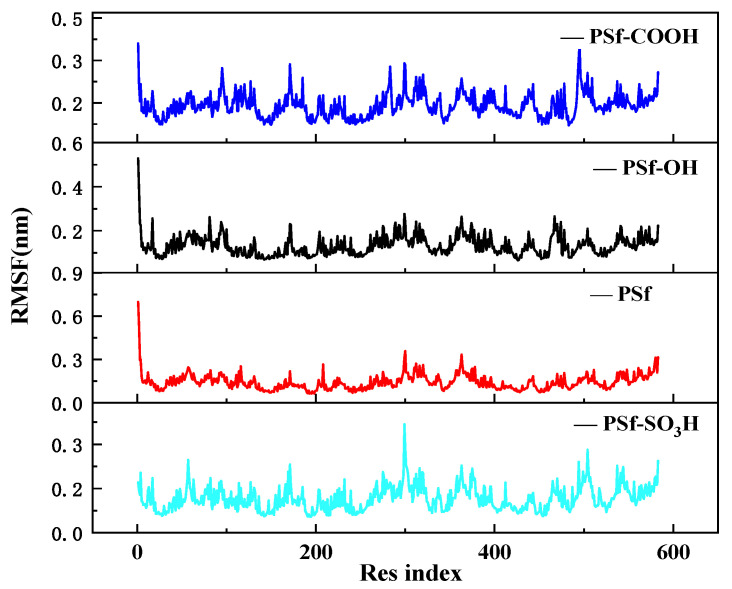
Simulation of RMSF for each residue in the final stage of BSA.

**Figure 7 polymers-16-01637-f007:**
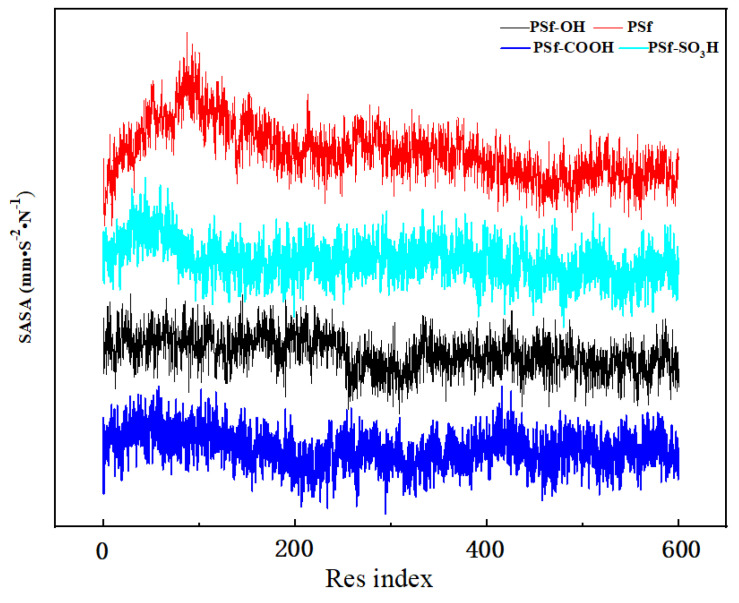
Illustration of the simulated SASA of each residue in the final stage of BSA.

**Figure 8 polymers-16-01637-f008:**
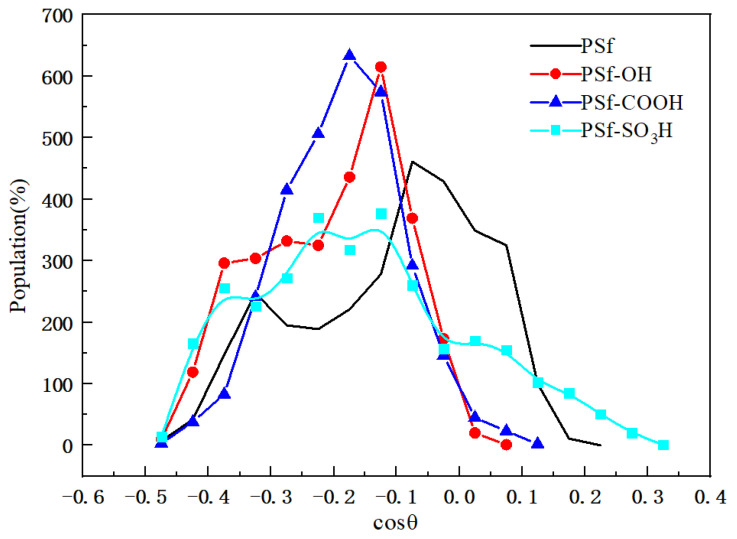
The cosine distribution curve of the orientation angle of BSA molecules in the final stage simulation.

**Figure 9 polymers-16-01637-f009:**
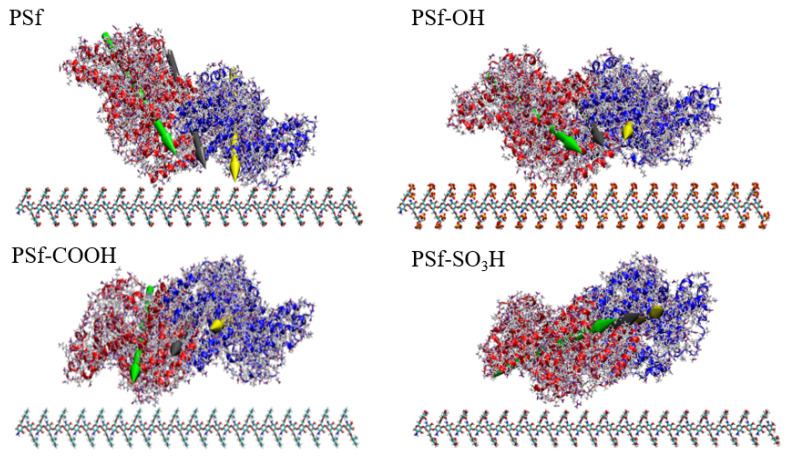
Binding sites of the BSA at the end of the simulation.

**Figure 10 polymers-16-01637-f010:**
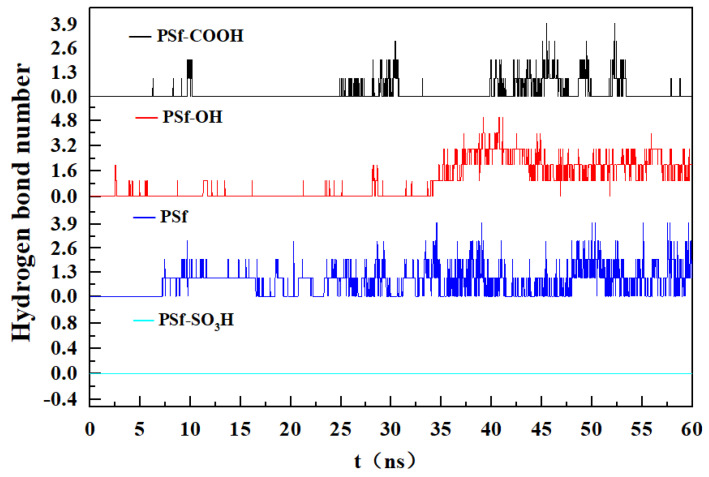
The number of hydrogen bonds between BSA and the membrane surface.

**Table 1 polymers-16-01637-t001:** The elemental surface compositions of PSf, PSf-OH, PSf-COOH, and PSf-SO_3_H.

Samples	Elemental (at%)			
	O%	N%	C%	S%
PSf	20.54	0.12	76.83	2.51
PSf-OH	21.93	0.14	75.73	2.20
PSf-COOH	23.99	0.23	73.02	2.75
PSf-SO_3_H	26.41	3.29	65.29	4.99

**Table 2 polymers-16-01637-t002:** Interaction energy of different graft membrane–BSA interface.

Interaction Interface	$E_{vdw}$	$E_{elect}$
PSf-CH_3_-BSA (kJ·mol^−1^)	−36	−270
PSf-OH-BSA (kJ·mol^−1^)	−108	−330
PSf-COOH-BSA (kJ·mol^−1^)	−156	−480
PSf-SO_3_H-BSA (kJ·mol^−1^)	−253	−253

## Data Availability

Data are contained within the article.

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
