# Peer review of "Interfacial Interaction between Functionalization of Polysulfone Membrane Materials and Protein Adsorption"

_polymers, 2024, doi:10.3390/polym16121637_

Round 1

Reviewer 1 Report (Previous Reviewer 1)

Comments and Suggestions for Authors

The author has appropriately addressed all the concerns raised during the previous review. The modifications and clarifications added have significantly improved the clarity and quality of the manuscript.

However, the following points should be addressed before publication:

1. In line 123, the authors mention using the CASTEP tool. However, reference 20 pertains to CASTEP, not reference 21.
2. In Figure 9, it is important to ensure that all simulation snapshots are captured under the same scale to keep consistency and comparability. Additionally, the red underline under 'Psf' in the title of each snapshot should be removed.

Comments on the Quality of English Language

The manuscript requires an English polish to be more native. It contains minor grammatical errors and inconsistencies, such as the article usage and typos.

Author Response

Dear reviewers,

Many thanks for your attention on our manuscript entitled “Mechanism of Anti-Protein Adsorption through Interfacial Functionalization of Polysulfone Membrane Materials” (ID: polymers-3043498). Once again, we appreciate reviewers’ useful comments and suggestions. The reviewers’ comments are very helpful. We have carefully revised the manuscript according to the reviewers’ comments, and marked with red color in manuscript. The following is a point-to-point response to the reviewers’ comments. All the comments are marked with red color.

Reviewer 2 Report (Previous Reviewer 2)

Comments and Suggestions for Authors

Compared to the previous version of the article, the authors have improved the text of the manuscript, but a number of serious comments remain.

1. Strictly speaking, the authors do not study the mechanisms of BSA adsorption, therefore, the article should probably be called “Interfacial Functionalization of Polysulfone Membrane Materials to decrease of Protein Adsorption.”

2. The authors did not explain why the dimer was chosen as a model rather than the BSA monomer, because in the blood, this albumin is found in monomeric form.

The authors completely ignored my comments from the previous review.

3. The authors should substantiate in more detail their conclusions about the type interactions during the adsorption of BSA on the PSf-OS3H surface. For this purpose, it is possible to visualize the interaction sites between the protein and the carrier for its adsorption in the LigPlot and Plip programs, i.e. identify putative amino acid residues and the types of bonds and interactions that are formed.

Docking for this type of study would be preferable, than MD, at least as the first stage: assembly of the complex of the carrier and the protein being studied. It is unclear why this was not done.

4. It is desirable to carry out experiments on the desorption of BSA in a concentration gradient of ammonium sulfate to identify electrostatic interactions and in a concentration gradient of Triton X-100 (or analogue) to identify so-called hydrophobic effects. Without experimental confirmation, any models are partly speculative.

Without a response to these comments, conclusions about the type of interactions to be premature.

Author Response

Dear reviewers,

Many thanks for your attention on our manuscript entitled “Mechanism of Anti-Protein Adsorption through Interfacial Functionalization of Polysulfone Membrane Materials(ID: polymers-3043498). Once again, we appreciate reviewers’ useful comments and suggestions. The reviewers’ comments are very helpful. We have carefully revised the manuscript according to the reviewers’ comments, and marked with red color in manuscript. The following is a point-to-point response to the reviewers’ comments. All the comments are marked with red color.

Reviewer 3 Report (New Reviewer)

Comments and Suggestions for Authors

Line 26: please organize the keywords alphabetically,

Line 68-71:  In the section “2.1 Materials” are listed various chemical compounds without any short description. Please make an introduction to the subject.

Line 80: as a temperature unit, please use the Kelvin units in the whole text,

Line 75-92 - organization of paragraph “2.1 Materials” should be changed. The division into a lot of subsections containing almost single sentences is unnecessary. Please connect the whole subsection in the one.

Line 108: “PSf-SO3H These” -  a dot is missing

Line 262: Figure 7 - Illegible chart. If possible, please change the presentation method. Figure 7 could be presented on a different scale to improve its resolution.

In the future studies, more attention could be dedicated to comparing the effects related to the hydrophobicity of the surface.

Author Response

Dear reviewers,

Many thanks for your attention on our manuscript entitled “Mechanism of Anti-Protein Adsorption through Interfacial Functionalization of Polysulfone Membrane Materials” (ID: polymers-3043498). We appreciate reviewers’ useful comments and suggestions. The reviewers’ comments are very helpful. We have carefully revised the manuscript according to the reviewers’ comments, and marked with red color in manuscript. The following is a point-to-point response to the reviewers’ comments. All the comments are marked with red color.

Round 2

Reviewer 2 Report (Previous Reviewer 2)

Comments and Suggestions for Authors

Authors took into account most of my comments. I recommend this article for publication.

This manuscript is a resubmission of an earlier submission. The following is a list of the peer review reports and author responses from that submission.

Round 1

Reviewer 1 Report

Comments and Suggestions for Authors

In the manuscript titled “Mechanism of Anti-Protein Adsorption through Interfacial Functionalization of Polysulfone Membrane Materials,” the authors studied the adsorption mechanisms of proteins on modified polysulfone (PSf) membrane surfaces through both experimental methods and molecular dynamics simulations. The authors started by modifying the surface properties of polysulfone(PSf) membranes with different functional end groups to investigate their influence on the absorption of bovine serum albumin (BSA). Moreover, the authors provided molecular dynamics simulations to offer insights into the adsorption mechanism, which has not been fully explained by previous studies.

However, the following points should be addressed before publication:

1. The authors described the methods for constructing the simulation model, but the simulation details, such as the boundary conditions and surface size, were not included. Since the model mimics the process of protein absorption, it is crucial to know whether the surface model was configured to prevent any unphysical movements.

2. In Figure 5, the simulation results of RMSD for different systems don’t show significant differences compared to the experimental characterization. Does this suggest that the simulation setup or RMSD measurements are insufficient to capture the adsorption differences?

3. In Figure 6, the y-axis, which represents the RMSF in nanometers, lacks numerical values to indicate the differences.

4. As the author mentioned in Figure 7, the SASA is a key parameter describing hydrophobicity. How was this data measured in simulations? Additionally, due to the fluctuations of the measurement, the results across the four systems don’t show distinguishable results either.

Comments on the Quality of English Language

The manuscript’s language and clarity could be improved to meet the journal’s standards. For example, ensure consistency in terms of initialism, such as “RMSF” and “RMS fluctuation.” Each abbreviation should be fully defined upon its first use. Some sentences need to be rephrased.

Author Response

Dear reviewers,

Many thanks for your attention on our manuscript entitled “Mechanism of Anti-Protein Adsorption through Interfacial Functionalization of Polysulfone Membrane Materials” (ID: polymers-3017332). We appreciate reviewers’ useful comments and suggestions. The reviewers’ comments are very helpful. We have carefully revised the manuscript according to the reviewers’ comments, and marked with red color in manuscript. The following is a point-to-point response to the reviewers’ comments. All the comments are marked with red color.

Reviewer 2 Report

Comments and Suggestions for Authors

The article is devoted to actual topic and has high applied significance. However, the manuscript raises a number of questions.

1. The essence of the term “anti-protein” is not fully understood. In the article, it is declared almost as an object of research, plus it is mentioned in the mysterious phrase “This provides a theoretical basis for surface modification of PSf membranes and improving their anti-protein properties.” What is meant by anti-protein properties?

2. At the end of the Introduction section, the authors write that “This paper utilizes Molecular Dynamics (MD) method to analyze protein adsorption on membrane surfaces from a molecular perspective” and “This provides a theoretical basis for surface modification of PSf membranes and improving their anti-protein properties". Because in addition to calculations, the article also contains the results of real laboratory experiments; at the end of the Introduction section they should be mentioned and the purpose of the work should be more correctly formulated.

3. The authors did not indicate in what software the structures of their membranes were drawn and in what force field the charges were placed on the surface of the models.

4. In addition, it is necessary to justify the choice of the dimer rather than the BSA monomer as a model.

5. The authors should substantiate in more detail their conclusions about the electrostatic interactions during the adsorption of BSA on the PSf-OS3H surface and the hydrophobic interactions between the protein and the membrane. For this purpose, it is possible to visualize the interaction sites between the protein and the carrier for its adsorption in the LigPlot and Plip programs, i.e. identify putative amino acid residues and the types of bonds and interactions that are formed.

Docking for this type of study would be preferable, at least as the first stage: assembly of the complex of the carrier and the protein being studied. Then MD, of course, shows the process of dynamic interaction.

The power of both computers and MD packages is such that they make it possible to count entire layers of membranes. It is unclear why this was not done.

6. In addition, it is desirable to carry out experiments on the desorption of BSA in a concentration gradient of ammonium sulfate to identify electrostatic interactions and in a concentration gradient of Triton X-100 (or analogue) to identify so-called hydrophobic interactions. By the way, it is more correct to talk not about hydrophobic, but about the totality of London and van der Waals interactions. Without the above, I consider the authors’ conclusions about electrostatic and hydrophobic interactions to be premature.

7. The contribution of the solvent is not clear. Was the dynamics process calculated in an aqueous environment? At what pH? What experimental data is there on this matter?

It is necessary to take into account the influence of pH and composition of buffers in which BSA adsorption is planned, because these factors will significantly influence the types of connections and interactions formed during the formation of the complex.

8. The phrase is unclear: the probability of irreversible folding and deformation during the adsorption process is small. — Are there any experimental data indicating incorrect folding or, in general, some conformational changes in the protein during adsorption?

Author Response

(The authors gave the same response as above.)
